# The Potential of Fish Oil Components and Manuka Honey in Tackling Chronic Wound Treatment

**DOI:** 10.3390/microorganisms12081593

**Published:** 2024-08-05

**Authors:** Jenna Clare, Martin R. Lindley, Elizabeth Ratcliffe

**Affiliations:** 1Department of Chemical Engineering, Loughborough University, Loughborough LE11 3TU, UK; 2School of Health Sciences, Faculty of Medicine and Health, University of New South Wales, Sydney 2052, Australia; m.lindley@unsw.edu.au

**Keywords:** docosahexaenoic acid, eicosapentaenoic acid, resolvins, biofilm, ESKAPE, manuka honey, wound

## Abstract

Chronic wounds are becoming an increasing burden on healthcare services, as they have extended healing times and are susceptible to infection, with many failing to heal, which can lead ultimately to amputation. Due to the additional rise in antimicrobial resistance and emergence of difficult-to-treat *Enterococcus faecium*, *Staphylococcus aureus*, *Klebsiella pneumoniae*, *Acinetobacter baumannii*, *Pseudomonas aeruginosa*, and *Enterobacter* spp. (ESKAPE pathogens), novel treatments will soon be required asides from traditional antibiotics. Many natural substances have been identified as having the potential to aid in both preventing infection and increasing the speed of wound closure processes. Manuka honey is already in some cases used as a topical treatment in the form of ointments, which in conjunction with dressings and fish skin grafts are an existing US Food and Drug Administration-approved treatment option. These existing treatment options indicate that fatty acids from fish oil and manuka honey are well tolerated by the body, and if the active components of the treatments were better understood, they could make valuable additions to topical treatment options. This review considers two prominent natural substances with established manufacturing and global distribution—marine based fatty acids (including their metabolites) and manuka honey—their function as antimicrobials and how they can aid in wound repair, two important aspects leading to resolution of chronic wounds.

## 1. Introduction

Surface wounds are a common issue and are classified as either acute or chronic, with further subclassifications dividing them into pressure ulcers, diabetic foot ulcers (DFUs), traumatic wounds, and surgical wounds [1]. In all these instances the skin is disrupted, breaking open the immune system’s first line of defence and leaving the body vulnerable to pathogens that may enter [2]. Often wounds heal well, but they can be impeded by excessive inflammation and infections. Chronic wounds are also becoming increasingly more common worldwide, with an estimated 15% of over-70s living with one [3]. These more serious wounds persist for an average of 12–13 months, and up to 70% of patients report reoccurrence of the wound [4]. There is a lack of definition as to how long a wound should normally take to heal as too many factors are present in the process, including wound size, type, and the patient’s age, diet, or other health complications [5]. There are many types of wounds, such as surgically induced cuts and burns, but diabetic foot ulcers are a particular example of potentially chronic wounds, as they are often impeded by other complications caused by diabetes, such as lack of blood flow to the extremities and a lack of growth factors at the site of injury [6]. Due to the increasing number of cases of diabetes across Europe and worldwide [7], diabetic foot ulcers are fast becoming a large burden on many healthcare services and so effective treatments for chronic wounds must be identified.

Wound healing is split into four phases: haemostasis, inflammation, proliferation, and remodelling. In haemostasis, the body reacts to wound infliction by staunching the blood flow to minimise loss. This is achieved by activation of the clotting cascade via either the intrinsic or extrinsic pathways, or by activating platelets. In all of these scenarios, thrombin is activated and a clot will form, either by platelet aggregation or conversion of fibrinogen to fibrin [8]. Inflammation consists of two phases: the initiation and resolution phases. Inflammation can also be split into either chronic or acute categories, where acute inflammation is resolved in good time. If not, acute inflammation becomes chronic [9] and is often linked to a range of conditions including diabetes, asthma, rheumatoid arthritis, Alzheimer’s disease, and cardiovascular disease [10].

Wound healing is the process by which skin is restored after injury and has several important aspects involved for it to happen unimpeded. One of these is the ability to keep the area clean and free of pathogens to give the body the chance to heal without being slowed by the need to also fight infection [2]. Untreated infections can progress and become biofilms, where the bacteria create a community under a slimy layer wherein they have greater communication and nutrient sharing abilities, as well as being afforded extra protection by the slime layer [11]. It is also important that the wound closes in suitable time to avoid the wound becoming chronic. Wound closure also restores the surface of the skin, which acts as a natural barrier to invading pathogens. Although the body is well suited to regenerating cells and healing in its own way, certain substances can boost these processes and can lead to faster wound closure [5].

Many natural substances have been identified as having the potential to aid in both these processes [12,13], and characterizing substances that can provide the benefit of both preventing infection and increasing the speed of wound closure may help advances in wound care and development of new treatments. This is especially important with the growing issue of antimicrobial resistance, as antibiotics are becoming increasingly ineffective against common pathogens that infect wounds and cause biofilms. Identification of novel broad-spectrum antimicrobials would be highly beneficial, as research into new antibiotics has slowed considerably and new treatments are highly sought after. Broad-spectrum antibiotics can treat many species of bacteria and can be used to treat biofilm infections. A treatment that could both provide an antimicrobial effect and aid in decreasing the time of wound closure would be highly advantageous in treating chronic wounds, particularly if it was low in cost.

Two of the most prominent natural substances that are already produced as food-grade ingredients and medical grade products with established manufacturing and global distribution identified as having medicinal properties are fish oil [14] and manuka honey [15]. Both have been widely used for a long time throughout history to aid in wound healing and a variety of other conditions, but it was not until more recently that scientific research delved into the components of these to determine what it is about them that provides medicinal effects and the mechanisms of actions that are involved that can aid wound healing. Use of natural substances as potential complementary topical treatments could infer advantages such as reduced progression of drug resistance and the benefit of not disrupting the gut microbiome during the healing stage [16]. Topical treatments can be used throughout wound healing. Some are designed to be prophylactic and avoid wound contamination, which could aid in prevention of chronic wound formation.

There are multiple formats a topical treatment can take, including direct application, gels, and ointments, and these substances can also be incorporated into smart dressings. Manuka honey is already in some cases used as a topical treatment in the form of ointments and in conjunction with dressings, although clinical trials examining its use have often shown to be of too low a quality to draw valid conclusions [17]. Fish skin grafts are an existing US Food and Drug Administration (FDA)-approved treatment option, though these are sometimes seen as less favourable due to their xenogeneic origin [18,19]. These existing treatment options indicate that fish oil fatty acids and manuka honey are well tolerated by the body and that if the active components of the treatments were better understood, could make valuable additions to topical treatment options. This review aims to assess innovative solutions for wound healing, particularly the fatty acid components metabolised from fish oils for these abilities alongside the better-understood manuka honey, to understand the potential benefits, contradictions, and the need for consistency in future research.

## 2. Therapeutic Fish Oil

Fish oil has been used therapeutically for centuries as it is low cost and readily available. Its first recorded mentions were from Hippocrates and Pliny, who noted its medicinal benefits and used it on the skin as well as consuming it [14]. It is also thought that the poorer population were using it for some time before clinicians later became aware of it in the 18th and 19th centuries, where it was prescribed to rheumatic patients to rub into their joints, producing some ‘astonishing’ results, further leading to its use in the treatment of gout, rickets, and tuberculosis throughout Europe [14]. In the 20th century, paediatricians began to recommend that children should receive it in regular doses for their general health as a preventative action [14]. The global omega 3 supplements market size was valued at USD 5.58 billion in 2020 and is expected to expand at a compound annual growth rate (CAGR) of 8.6% from 2020 to 2028 [20]. Its popularity has arisen due to readily available online claims that it can aid in preventing heart disease, arthritis [21], high blood pressure, and high cholesterol [22], issues that are commonplace and of concern among the general population.

### 2.1. Fish Oil Components

Omega fatty acids are available from a range of dietary sources, including fish, nuts, and plants. Omega fatty acids are comprised of polyunsaturated long chain fatty acids (PUFAs). Fatty acids are commonplace throughout the diet and are available in different formats. Unsaturated fatty acids from the diet become constituents of cell membranes and maintain different metabolic actions depending upon their structure. Omega-6 polyunsaturated fatty acids (PUFAs), arachidonic acid (AA), and omega-3 PUFAs α-linoleic acid (ALA), arachidonic acid (AA), docosahexaenoic acid (DHA), and eicosapentaenoic acid (EPA) are the most abundant and metabolically active [23]. Although PUFAs can be found from many dietary sources, marine life is particularly rich in DHA and EPA compared to plant sources, which are usually richer in ALA [24]. As such, EPA and DHA are often termed marine based fatty acids. From the marine-based fatty acids, the human body synthesizes resolvins (Rvs). E-series resolvins (denoted RvEs) are derived from EPA, and D-series (denoted as RvDs) are derived from DHA. There are multiple resolvins that have been identified from each series, and they are produced during the resolution stage of inflammation, which is where they gain their name from. Each series of resolvins is synthesized in a different pathway, which can be altered by the presence of aspirin [25].

DHA is converted to 17S-hydroperoxy-DHA by 15-lipoxygenase, and this intermediate is converted into one of six identified resolvins. Resolvins D1 through to D4 are shown in Figure 1B. EPA is converted by cyclooxygenase 2 (COX-2) into 18R-hydroxyeicosapentaenoic acid. This intermediate is also converted into one of four resolvins via an epoxidation reaction. RvE1 and RvE2 pathways are shown in Figure 1A. Some nonsteroidal anti-inflammatory drugs target the COX-2 pathway and produce similar aspirin-triggered resolvins [23]. EPA also produces prostaglandins and leukotrienes from different reactions, which also have roles in resolution. Leukotrienes inhibit leucocyte chemotaxis and function as a signal to stop inflammation [26]. DHA also has other metabolites besides resolvins, including protectins and maresins [27]. Only resolvins are produced by both EPA and DHA, and many of these molecules have been shown to aid in resolution of inflammation.

### 2.2. Fish Oil in Wound Healing and Resolution

It has been established that fish oil components have anti-inflammatory effects and may have a positive effect on decreasing wound healing time and keeping infection at bay. DHA-derived molecules have shown to be elevated during the healing process. Mice were treated for induced corneal injury with topical application of leukotriene A (LXA) or protectin 1 (PD1) three times a day for 96 h [28]. Compared to a control group, the test group had a faster rate of re-epithelialisation, and analysis of the cornea showed that even when the treatments were not applied, there were raised levels of the molecules present. Interestingly EPA-derived molecules had no effect on corneal healing [28]. Other murine studies also demonstrate the effects of PD1. Levy et al. (2007) [29] administered PD1 to mice before inducing a challenge to the airway to simulate allergic inflammation. The mice had reduced eosinophils shown histologically and less mucus in the airways. Measured exudate also suggested that some PD1 was produced as a natural response to inflammation. as it was found to be present, although in significantly smaller amounts. than in the treated mice.

Ruiz et al. (2019) [30] also suggest that both resolvins and maresins have an antimicrobial effect against *Mycobacterium tuberculosis*. Macrophages were infected with *M. tuberculosis*, then later subjected to treatments of RvD1, RvD2, PD1, LXA and maresin 1 (Mar1). The cytokines present after 24 h were measured, and it was shown that all treatments significantly regulated production of tumour necrosis factor alpha (TNF-α), which was present in decreased amounts (*p* < 0.05). The authors also observed that RvD1 and Mar1 led to enhanced antimicrobial activity and that the rate of cell death was significantly increased (RvD1 *p* < 0.05, Mar1 *p* < 0.01). The data revealed that although both molecules reduced bacterial titre, Mar1 showed a much more consistent reduction than RvD1. In addition, RvD2 reduced the average count of viable bacteria, but failed to do this consistently and was not statistically significant [30].

Resolvins (Rvs) are produced at different times during the inflammatory process, as levels of RvD3 rise much later than RvD2 [31]. Previous research shows RvD2 increases the most rapidly after initiation of inflammation. It peaks first and rapidly declines as the other D-series resolvins are still climbing. RvD1, RvD2, and RvD3 all increase steadily to begin with, with RvD3 rising past these to peak after RvD2. Interestingly, the level of RvD3 stays high even after the other resolvins have decreased [32].

The exact actions of the resolvins are largely unknown [33], but many studies have observed different effects from them, and they do not all exert the same effect. RvD2 and RvD3 can boost phagocytosis of *Escherichia coli* (*E. coli*) in vitro, increasing bacterial clearance [34]. RvD1 and RvD4 also show an increase in phagocytosis, but to a significantly smaller degree (*p* < 0.01) [31]. RvD3 has also been shown to significantly stimulate efferocytosis where the cellular debris and apoptotic cells are cleared before return to homeostasis [32].

Resolvins are also effective in reducing cytokines including interleukins IL-6 [32] and IL-10 [35], tumour necrosis factor TNF-α [36], and IL-1β [37]. This reduction occurs during the resolution stage of inflammation and ensures that the injury will not have too large a cytokine response, as unregulated cytokines can cause a wound to become chronic [38]. It has been established that the resolvins have different receptors that may lead to them having different effects, as when RvE1 and RvE2 are administered together in animal models of inflammation, their observed effect is additive [39].

Aside from chronic wounds, fatty acids have already been suggested as topical treatments for other skin conditions, including acne and superficial infections, due to their anti-inflammatory effects [16]. This shows that they are tolerated as topical treatments, and this has the potential to translate into remedies for other surface ailments. Other needs for topical treatments are the killing of methicillin-resistant *Staphylococcus aureus* (MRSA) in the nasal passages that is often performed prior to surgery [16], showing that although this review is based on theoretical chronic wounds, there are other important cases where topical antimicrobials may be required.

### 2.3. Fatty Acids and Bacterial Cells

Bacteria synthesize a range of fatty acid components through two main pathways, known as the fatty acid synthesis pathways, type I and II (FASI and FASII). Fatty acids are important for bacteria structure, as they are major components of the cell membrane and necessary as metabolic intermediates. FASI produces palmitates, while FASII produces a much wider range of molecules [40]. Bacteria can acquire fatty acids exogenously to facilitate these reactions. Gram-positive and -negative bacteria have different genes associated with this, which are fatty acid kinase FakAB and fatty acid transport protein FadL, respectively [41].

In Gram-negative bacteria, FASII produces acyl–acyl carrier protein (ACP) and β-hydroxyacyl-ACP, molecules that become essential parts of the membrane. Gram-positive bacteria use FASII-derived acids for membrane phospholipid and glycolipid formation. For Gram-negative bacteria, *E. coli* was considered a standard representative bacterium for a long time, which is why many of the existing antimicrobial studies have used it; therefore, it was the first bacterium for which the FASII pathway was properly determined [42].

However, the authors of the review suggest that *E. coli* is not an appropriate representative Gram-negative bacterium. Gram-negative bacteria cannot convert fatty acids into acyl–acyl coenzyme A reductase (ACR). Exogenous fatty acids cannot enter the FASII pathway, although no reasons for this difference have been suggested [42].

As an important part of bacterial health, the FASII pathway has become of interest due to its potential to develop antibiotics that target the pathways involved. In the hope that this can kill bacteria [43], it has also been suggested that targeting of these pathways will have no toxic effects on humans [40]. This area of research is relatively new; however, there is already evidence that *S. aureus* can develop resistance mechanisms against these treatments [43,44].

Several authors have demonstrated that uptake of exogenous fatty acids can lead to an increase in bacterial growth compared to when bacteria have no external sources of fatty acids [44,45]. This has been documented in both Gram-positive and -negative bacteria, including with serum fatty acids and *S. aureus*, *Lactobacillus* spp. [44] and *K. pneumoniae* when exposed to DHA and EPA [44]. Adams et al. (2021) [41] infected mice with *Acinetobacter baumanii*. One group of mice were wild type. The other were deficient in fadL, so would not take up any exogenous fatty acids. Plasma analysis before and after infection showed that fatty acids in plasma had fallen, suggesting that they had been taken up by the bacteria. Colony-forming unit (CFU) counts showed that the fadL-deficient mice had significantly fewer bacteria present in all tissues assessed (*p* < 0.001), including bronchoalveolar lumen, lung, pleural cavity, liver, and spleen.

Herndon et al. (2020) [46] demonstrated that *E. coli* did not incorporate DHA into its membrane. At the same position as the other PUFAs assessed, DHA also caused a 25% increase in cell permeability. They also investigated biofilm formation and found that DHA supplementation of minimal media led to a doubling of biofilm content when compared to a control. It is difficult to draw conclusions here, due to the measurement system used. The authors used a plate reader with crystal violet, which does not provide an accurate cell count and instead stains the entire biofilm mass, and no cell counts were provided. This leaves ambiguity when determining to what extent the viable cells have been eradicated. However, despite this, regarding FASII and exogenous fatty acid uptake, there are many studies that demonstrate reduction in viable bacteria cells when exposed to polyunsaturated fatty acids. Herndon et al. (2020) [46] and Hobby et al. (2019) [45] both state that DHA increases membrane permeability, and this is usually associated with an increase in lysis as it leaves the cell more vulnerable to antibiotics and other harmful substances. Both authors used M9 minimal medium with 0.4% casamino acids. This medium contains no carbon source, so the addition of the PUFAs are the only available sources. The use of Müller–Hinton broth is considered the standard for antimicrobial susceptibility testing, as outlined by the European Committee on Antimicrobial Susceptibility Testing (EUCAST) [47]. Due to its content, it is considered more representative of a mammalian biological system compared to the salts that make up M9 media, as it contains extracellular matrix components. However, the two studies were not specifically testing antimicrobial susceptibility and limited their carbon sources to measure uptake of PUFAs without interference from other substances.

Le and Desbois (2017) [48] designed their study to investigate the effect of EPA on the bacterial membrane, and assessed the cell leakage that they hypothesized was a cause of EPA incorporation into the membrane. *S. aureus* and *Bacillus cereus* were exposed to a range of EPA concentrations for 30 min. CFU/mL was established, a portion of the sample was centrifuged, and the supernatant passed through a 0.22 μm filter. The absorbance of the filtered sample was measured at 260 nm and compared to a control, which was taken and prepared in the same manner at time point 0. A 260 nm wavelength is used to measure deoxyribonucleic acid (DNA) and ribonucleic acid (RNA), metabolites, and ions released from cells [49], although it is unclear from the analysis used which components were present. The study aimed to use the absorbance measurements to assess the quantity of cell contents that had leaked due to the increase in membrane permeability, and found that for *S. aureus* at 64 mg/L and higher, there was more material present in the filtrate than in the control. Although the mean had increased, there was no statistical analysis provided. *B. cereus* had an increase in measured cell material at lower concentrations. It became detectable at 32 mg/L and had increased to 64 mg/L, where absorbance was around 0.1. The decrease in viable CFU was correlated with an increase in absorbing material, and it was suggested that this could show membrane disruption and cell lysis from EPA exposure.

Fatty acids can also influence reducing bacterial growth. Where naturally present on the skin, they make the environment acidic to make growth unfavourable, and their presence can also up- or downregulate expression of some toxins, enzymes, and haemolysins [50]. It is also thought that direct antibacterial activity is related to fatty acid interaction with the cell membrane, with multiple processes likely being involved at a time. These include solubilization of the membrane, disruption of the electron transport chain, enzyme inhibition, interference with oxidative phosphorylation, and inhibition of nutrient uptake [50].

These differences in the literature show a need for proper understanding of how specific PUFAs interact with bacterial cells, as there is a range of papers reporting improvement to bacterial growth, while others report death and inhibition. It is important to establish what effects PUFAs can have on bacteria if they are considered antimicrobial treatments.

### 2.4. Fatty Acids and Mammalian Cells

Similarly to bacteria, mammalian cells also incorporate PUFAs into their membranes, which are comprised of a phospholipid bilayer that makes up the plasma membrane. This bilayer also contains a range of proteins that can move around, and the membrane is hence described as a ‘fluid mosaic’ [51]. ALA, DHA and EPA are the major PUFAs found within the membrane [52]. PUFAs in the membrane are responsible for the fluidity and integrity of the membrane [53] and the structure of the molecule is important, as the impact on the membrane is determined by the chain length, the degree of saturation, and the placement of the double bonds. The more unsaturated a chain, the more fluid the membrane becomes [54]. Membrane fluidity is considered advantageous, as flexibility of the cell is necessary for environmental adaptation and aids in cellular division and differentiation [55]. Increasing membrane fluidity could therefore aid in wound healing if division were encouraged, as this would produce more cells as needed to re-cover the injured area.

However, copious quantities of PUFAs can be toxic to cells, as once the membrane is saturated, the PUFAs accumulate within the cell and form triglycerides. While triglycerides are not toxic, if PUFAs are not converted, they can lead to toxicity [53]. It is also thought that elevated levels of PUFAs in the membrane would increase the susceptibility of the membrane to peroxidation, which results in the generation of lipid peroxides [56].

DHA and EPA are known to behave differently when incorporated into mammalian cells. DHA causes a larger increase in membrane fluidity. This is suspected to be because of its longer chain and extra double bond compared to EPA [56,57].

### 2.5. Fish Oil Antimicrobial Activity

Over the last few decades, research has steadily increased on marine-based fatty acids (Figure 2) but is still low, especially compared to other emerging antimicrobial alternatives to antibiotics, such as bacteriophage therapy. Whereas fish oils have the potential for antimicrobial efficacy against multiple pathogenic organisms, bacteriophages are viruses that infect bacteria, and bacteriophage therapy uses specific phage viruses that target and kill specific pathogenic bacterial strains. Journal articles concerning bacteriophages have remained high over the past few decades, with investigations into therapeutic phage cocktails and non-therapeutic applications. In 2019, there were 1504 papers published concerning bacteriophage technology, and 804 regarding the broader-spectrum EPA.

Some in vitro studies have shown that EPA can have an antimicrobial effect against a range of organisms, including *S. aureus*, *K. pneumoniae*, and *E. faecalis* [58]. EPA (concentration 128 mg/L) can eradicate planktonic cultures of *S. aureus* and *Bacillius cereus* cultures in the exponential phase when at a concentration of 5 × 10^5^ CFU/mL. This was demonstrated with standard minimum inhibitory concentration assays in well plates. The bacteria did not become resistant over 13 passages when they were subcultured in solutions containing low levels of EPA either [48]. This is of significant interest, as a treatment that could be used repeatedly without eventually becoming ineffective would be of significant use in the post-antibiotic era. This study had no information on DHA, and it would be of interest to see if similar effects were observed.

Much of the research concerning the application of marine-based fatty acid components to biofilms is focused on the eradication of bacteria that cause issues, such as dental biofilms. *Porphyromonas gingivalis* and *Fusobacterium nucleatum* are two bacteria that commonly form biofilms on the teeth [13]. Authors have observed that both DHA and EPA inhibited growth of *P. gingivalis*, but had a much smaller effect on *F. nucleatum*, which had a significantly reduced growth rate when treated with 100 μM, but after 48 h had still achieved an optical density (OD) equal to the untreated control. The authors notate this as a minimum inhibitory concentration (MIC) of >100 μM, although their MIC was evaluated after the total 48 h, and visual analysis of the graph suggests that after 18 h, there had been no significant growth after 100 μM of EPA treatment, although no statistical analysis was provided. This effect was not seen for treatment of *F. nucleatum* with DHA, where all treatment concentrations showed a large amount of growth after 18 h.

Sun et al. (2017) [59] subjected *F. nucleatum* and *P. gingivalis* to treatments of either EPA or DHA. Both treatments at 100 μM proved capable of significantly reducing the species as planktonic bacteria (*p* < 0.01) and inhibiting growth of the organism when the treatment was given at the point of inoculation. The study also subjected biofilms that had grown in 96-well plates for 24 h previously to 100 μM of either EPA or DHA for 24 h. Cell viability was measured using Syto9 and propidium iodide, which stains intact cells and those with damaged membranes in distinct colours. Analysis of the stains showed that EPA had a stronger effect in both reducing the thickness of the biofilm (*p* < 0.001) than DHA (*p* < 0.01) and in damaging bacteria [59]. Although both had a significance of *p* < 0.001, the data show that EPA damaged approximately 62% of cells and DHA 58%. This later study includes scope on treating biofilms that have already formed and provides insight as to how EPA and DHA could act if preventative treatment was not available. Although the studies on dental biofilms are less relevant to treating wounds, studies by Sun et al. (2016, 2017) [13,59] do show that EPA and DHA can inhibit biofilm growth in some organisms and warrant further investigation, though there is no guarantee that this will translate to other pathogens.

Most of the existing studies apply in vivo methods to their investigations, and there are three main routes of administration that are used: dietary supplementation, topical application, and direct injection to the site. The in vivo studies build on the idea that fish oil supplements are commonplace and can in most cases be safely and easily consumed, so often do not investigate specific fatty acids or resolvins separately. Tihista and Echavarría (2018) [60] demonstrated that fish oil has great effects by itself, reducing the risk of sepsis and septic shock in burns patients receiving enteral nutrition. Patients admitted to a burns ward (n = 92) received enteral nutrition within 24 h of receiving a burn and received either a fish oil-rich diet or sunflower oil administered normally. The rate of severe sepsis and septic shock was significantly lower in the group with the fish oil diet (*p* = 0.03). It was also reported that other complications associated with burn care in hospital were lower in the treatment group (although not significantly). These conditions included thrombosis, thrombocytopenia, acute renal failure, and hyperglycaemia [60].

Like Sun et al. (2016) [13], the work by Codagnone et al. (2017) [61] also supports the theory of specificity in treating bacteria. Mice were injected with *P. aeruginosa* and RvD1 administered intragastrically under anaesthetic. The lungs were later homogenized and subjected to analysis by ELISA and liquid chromatography mass spectrometry. Bacterial counts were also performed by plating onto agar and calculating the CFU/mL. Interleukin IL-1β was significantly reduced, as was bacterial titre of *P. aeruginosa* (*p* < 0.05), which was reduced from 10^7^ to 10^5^ CFU [61]. The study was also repeated using RvE1 and found that this had no significant effect on the bacterial titre or the cytokines present, suggesting the molecules have different targets. RvD1 was also particularly of interest. as it was equally effective when administered at the height of infection as it was at the beginning. This is one area where many of the existing studies are lacking—many have pretreated murine models with fish oil components [36,62]. While this is still useful, it provides limited information as to how the treatment would translate to a clinical setting where patients cannot necessarily be pretreated.

Diabetic foot ulcer (DFU) patients have shown significant decreases in ulcer length and depth (*p* = 0.03 and *p* = 0.01) compared to a control group when given omega-3 supplements twice a day (for a daily total of 1000 mg) for 12 weeks, with a complete lack of side effects (n = 60) [63]. This was given in addition to regular antibiotic treatment, and as no microbial analysis was performed, the study gives no insight into the antimicrobial activity of omega 3, but still demonstrates that omega 3 can aid in wound healing. The study would also benefit from analysis of inflammatory markers to provide insight as to the effect of the fish oil at the molecular level [63].

However, the evidence for marine-based fatty acid derivatives aiding in wound healing is mixed. McDaniel et al. (2017) [64] gave patients (n = 40) with venous leg ulcers dietary supplements containing 2.5 g of EPA and 0.5 g of DHA each day for a total of 4 weeks. Polymorphonuclear leucocytes (PMNs) from blood samples were analysed using flow cytometry. Patients in the treatment group had a visibly better wound closure rate, but the results remained statistically non-significant (*p* = 0.09), and blood work showed no significant reduction in activated PMN compared to a placebo-treated control [64]. This could potentially hinder wound healing, as uncontrolled PMN activity can lead to destruction of new tissue and breakdown of much-needed growth factors [65]. McDaniel et al. [64] also considered the factor that fish oil is unappealing to many people, so their capsules were lemon-flavoured.

Studies performed in animals have also shown contrasting findings. Mice inflicted with wounds showed a lack of improved re-epithelialization when given fish oil rich in EPA and DHA compared to both an untreated group and a group administered olive oil after 14 days [66]. Treatments were administered 3 days before the creation of a 1 cm^2^ wound that was left uncovered and untreated and continued for another 14 days until the study’s end. After 7 days, those that had received the fish oil diet had significantly less wound closure than the groups receiving olive oil or water (*p* < 0.05). Even after 14 days, the water and olive oil groups had smaller wounds left (*p* < 0.05) [64]. Although the data presented in this study contradict others, it has strength in that they measured the wound closure over several time points.

Fish oil has had previous success when used as a topical application to treat wounds. Sasongko et al. (2018) [67] extracted oil from the snakehead fish and the eel (both found often in Indonesian water), which was refined to be used as an ointment. It was applied to rats with inflicted wounds: scratches were made along their backs that were 2 cm long and 2 mm deep. It was found that those that received the eel fish oil treatment had up to 92% wound closure, the snakehead fish oil gave a closure rate of 88%, and the untreated controls healed only 52% in the same time period [67]. The study could have benefited from investigating and reporting the quantity of DHA and EPA in each of their extracted oils, as information on what the oil contains could provide insight as to why one oil seemed more effective than the other.

Interestingly, Sasongko et al. (2018) [67] also trialled combinations of eel and snakehead fish oil in different ratios and found that while these were still more effective than the negative control (*p* < 0.05), they failed to be more effective than eel oil alone, even when the amount of oil used was equal to that of the eel oil used alone [67]. The difference, however, was not great, and without proper reporting on statistical values and variance displayed, it is difficult to conclude whether they had a significantly different effect. This study suggests that a topical application may be more effective than a dietary one, as demonstrated by the lack of closure in a study by Rosa et al. (2014) [66]. This may be because direct application ensures that the maximum amount of product possible interacts with the site, rather than becoming diluted throughout the body.

Naveh et al. (2011) [68] studied diabetic rats with chronic foot ulcers and used a topical administration of fish oil in their study. Diabetes was induced by injection with streptozotocin, and after diabetes was verified with a glucometer, 4 cm incisions were made on the dorsal skin. Either fish oil or corn oil (which contains omega 6 rather than the omega 3 in fish oil) was applied once daily with a dropper until the wound had healed, but the authors give no indication as to the quality of the oils or the fatty acid content. Wound closure was measured by tracing the wound onto graph paper and measuring the surface area by counting the squares. This is similar to how Rosa et al. (2014) [66] measured their wounds, but lacks digital analysis and may be less accurate if squares on the paper are only partially filled. The results are not presented well in this article: despite the methods stating that the percentage of wound healing was calculated, this was not actually displayed. The authors just report the areas of the wound measured with variance and the table lacks a lot of information including units of measurement and what the p values displayed within it correspond to. There are data of use later in the paper, where the authors report the time taken for full wound closure. The diabetic control group took on average 10 days longer for the wound to heal than a healthy but untreated control group. Diabetic rats treated topically with fish oil took significantly less time for wound closure (*p* < 0.01), but the corn oil was most effective, as the time for closure was similar to the healthy control group (*p* < 0.001 when compared to the untreated diabetic group). The results from both Naveh et al. (2011) [68] and Sasongko et al. (2018) [67] indicate that a topical application of fish oil components has a positive effect on the rate of healing.

From the range of studies found during the literature review, there is a wide scope of how fish oil components are being used and at what point in the metabolic pathway they are being applied. Studies focus on fish oil, EPA, and DHA, and then the resolvin substrates. There is also variety in the source of these products, as some use extracted fish oil [67] and may derive EPA and DHA from these. However, other studies (particularly including those that use resolvins) rely on chemically synthesized fish oil components, as the structure of the chemicals has been identified and they can be made in the lab. The synthesized versions of the chemicals may be more appropriate for clinical use, as this would not involve any animal products, which could incur ethical issues from some patients based on dietary or religious preferences.

A large amount of the existing research used animals in the methods, and with the lack of understanding of the mechanism of action by which the fish oil fatty acids work, the need for detailed in vitro studies is highlighted. This will bring understanding of the cellular response and could lead to more focused and thus valuable animal studies when the in vivo stage is reached. By investigating a wider panel of wound colonizing bacteria, it may provide insight into how the treatments are effective due to the differing nature and resistance mechanisms present in the species. When considering the wound healing aspect, analysis of the markers and cytokines involved in the response to the treatment can help aid in establishing their suitability for promoting wound healing.

## 3. Therapeutic Honey

Aside from fish oil, many other ‘natural remedies’ are gaining attention in scientific studies, and one of these is honey. Although its antimicrobial properties were first observed in 1892, honey has been used medicinally for centuries, with references dating back as far as the Old Testament of the Bible and the Quran [15]. It is often used to treat sore throats in patients with colds [69] and its use has been investigated in a variety of ways, including as an antimicrobial [70,71] and as a regenerative agent [12,72].

### 3.1. Honey Components and Classifications

Many of the studies mentioned above compare different varieties of honey, as it has been established that the composition of honey differs due to the local flora that the bees use to produce it [73]. Besides containing sugar and water, honey is also comprised of proteins, organic acids, vitamins (including B_6_, thiamine, and riboflavin), copper, iron and zinc. In total, honey contains over 200 substances [74].

Unlike with fish oil, the antimicrobial mechanisms of honey are better understood. Hydrogen peroxide is formed by oxidation of glucose, and this inhibits microbial growth through the production of reactive oxygen species, which damage DNA [75]. Glucose oxidation is a two-step redox reaction, which is accompanied by flavin adenine dinucleotide cofactor [76]. Phenols and organic acids (known as flavonoids) also provide an antimicrobial effect in the absence of hydrogen peroxide, showing non-peroxide activity, though the exact mechanism of this is unclear [77].

Bee defensin 1 has also been found in many types of honey and is secreted by bees during the manufacturing process. It has previously been shown to form pores in bacterial membranes in vitro and to have antibiofilm properties against *S. aureus* and *P. aeruginosa* [75,78]. The proposed mechanisms of action are summarized in Figure 3.

Manuka honey has gained popularity in recent years, and its proposal that methylglyoxal (MGO) is a vital component has led to its marketing based on MGO content. Manuka honey has unusually high levels of non-peroxide activity, and this is attributed to MGO content [82]. Manuka honey comes from New Zealand and Australia, as it is made by bees from the native manuka plant. It gains MGO through non-enzymatic conversion of dihydroxyacetone which is found in nectar [83]. There are other systems to rank manuka honey, such as the ‘Unique Manuka Factor’ (UMF), but more recently, the MGO rating has been widely adopted and is used most frequently. MGO 850+ is the highest-ranked honey that can be easily purchased commercially, although MGO ranks as low as MGO 40, with a large price difference between these.

To be considered a medical grade honey, it must be sterilised to rid the honey of bacteria and kill dormant bacterial endospores, with strict standards and regulations for quality, processing, and storage [84] followed. Despite its clear antibacterial properties, honey is not naturally sterile and can contain spores of *Clostridium botulinum*, hence the recommendation that honey should not be given to babies and young children to eat. Early studies reported that irradiating honey did not affect its antimicrobial properties, though the authors neglected to show the antimicrobial data in favour of the decrease of bacteria found in the honey [85]. Other authors have also investigated the effects of gamma radiation on honey, and their reported data did not show differences between the antimicrobial abilities of untreated and irradiated honey [86]. A paper presented by Molan and Allen (1996) [86] still lacks proper statistical analysis, but their standard deviation was small and the sample size is reasonable (n = 16). The reader is left to consider the results themselves. Both papers conclude that it is safe to sterilize honey from 25–50 kilograys (kGy), although it is reported that this reduces its viscosity by up to 50% [87].

Sterilisation may be achieved through different methods, including heat sterilization, which is commonly applied to food-grade honey and will kill bacteria, but many methods also inactivate the antimicrobial and pro-healing components of honey, whereas gamma irradiation and ozonation are the two sterilization techniques shown to preserve these properties, with gamma irradiation providing superior elimination of bacteria [88].

Medical grade honey must also lack other contaminants common in honey: herbicides, pesticides, and heavy metals are often identified in untreated honey [84]. Medical grade honey is not exclusively prepared from manuka honey and can be made from any honey as long as the requirements are met. In 2007, the FDA approved the first honey-based dressing that used medical grade honey, although it is still not necessarily used as the first treatment option, despite positive responses from patients [89].

Despite medicinal honey being easily purchased in chemists, there are no existing legal requirements for making it. A search of the Medicines and Healthcare Products Regulatory Agency (MHRA) does not show that Activon tube honey is registered as an official medicine, and it does not display a product license code as required for marketing medicines in the UK [90]. The National Institute for Health and Care Excellence (NICE) lists several brands that they recommend for use, including Activon, Medihoney, and Revamil [91]. This does not prevent other companies from marketing their own honey as medical grade, and without prior understanding of the labelling systems, there is nothing to prevent patients from ordering non-irradiated honey with potential infectious organisms present.

Beehives are often treated with antibiotics such as sulphonamides, tetracyclines, and macrolides to prevent disease in honeybees, and these residues can contaminate the honey produced [92]. In Europe, this is prohibited, and sale of such honey is restricted. Most European Union (EU) countries will not sell honey from antibiotic-treated hives. The UK has set out maximum residue levels (MRLs) for each class of antibiotic, so such honeys can be sold, but only if they have fewer than 10–50 parts per billion (ppb) [92].

Honey from other countries, including India, Portugal, and Spain, has also had reports of pesticide contamination. Due to the beneficial activity of bees, they are not classed as pests, and while bees themselves are not directly treated with pesticides, they can feed on pesticide-treated plants and transfer the pesticides to the honey produced [93]. Blasco et al. [93] tested market honey for a wide range of pesticides, and found that the most frequently detected were organochlorines (dichloro-diphenyl-trichloroethane (DDT), hexachlorocyclohexane (HCH), and hexachlorobenzene (HCB)) and carbamates.

Although organochlorines have been banned in Europe for many years, they persist in the soil and can contaminate plants that grow in previously treated areas. Manuka honey comes specifically from Australia, which also prohibited the use of organochlorines many years ago.

There is a lack of harmonized strict guidelines for standardized formulations of medical grade honey and ambiguity around adherence of honey-based wound care products to medical grade standard. Efforts made towards safer medical grade honey formulations effective for wound healing include the development of guidance on standards and criteria for collection, contamination, sterilization, production, storage, physiochemical, legal, and safety issues for medical grade honey [88].

### 3.2. Honey Antimicrobial Activity

Manuka honey has previously been used in many studies, and despite the recommendations in the UK that only medical grade honey be used for treatment, there is a wide variety of honeys that have been utilized in research. This includes medical grade sterilized honey, but also includes food-grade honey and honey obtained directly from apiaries. Research into manuka honey can be up to 20 years old, however, and the guidance was not put into place until later. The studies are also from a range of countries, who may not have the same guidance in place.

Treatment in vitro of MRSA on agar with honey at a concentration of 5–20% (*w*/*v*) can prevent the bacteria from multiplying and reduce the viable count by up to 45% [94]. Microscopic analysis also shows that the cells have significantly more septal components than untreated cells (*p* < 0.001), suggesting that the honey inhibits the stage of cell division as the cells have failed to separate and successfully multiply [94].

Using honey as a regenerative agent has been evaluated using in vitro culture of skin cells and has been used in scratch model analysis previously. Ranzato, Martinotti and Burlando (2013) [12] seeded dermal fibroblasts to 12-well plates and used a scratch wound model where a pipette tip was used to inflict damage on the culture. Concentrations (0.1%) of diverse types of honey were applied, and after 24 h, the width of the wounded space was measured using digitized images and the analysis software ImageJ (National Institutes of Health Image J software (Bethesda, MD, USA)) and compared to measurements taken prior to treatment. Manuka honey caused significantly better closure than the untreated control, but was significantly outperformed by both buckwheat and acacia honeys, which were also employed in this study [72]. The authors’ statistical analysis showed that the buckwheat and acacia honey were not different from each other and that these two were significantly different from the control and the manuka samples, with *p* < 0.01 used as an indication of significance.

Ranzato, Martinotti and Burlando (2012) [72] also replicated their study using keratinocytes. They use the same varieties of honey, but this time documented the strength of the manuka factor using the older UMF system. The manuka honey was UMF15+, which correlates to approximately 514 MGO [95]. The scratch wound assay and cell migration assay were performed in an identical manner. The keratinocytes did also respond to the treatments and there was less variation between the efficacy of the three types of honey. All were significantly different to the control, and the manuka and acacia honeys were statistically different from the buckwheat sample (*p* < 0.01), although a graph showed that they were only slightly more effective.

DFUs have already shown benefits to treatment using honey, both in vitro and in vivo [96]. Kateel et al. (2018) [97] took wound swabs from patients and isolated five strains each of *E. coli*, *P. aeruginosa*, and *S. aureus*. Sterile honey was used (though no note is made of its floral origin) in concentrations from 25% to 100%. Although the authors did provide the specific gravity, which is useful in comparing to other studies, they did not specify whether the dilutions were prepared in a weight-by-volume or volume-by-volume capacity, reducing cross-study comparability. Minimum inhibitory and bactericidal concentration assays were performed using 1 × 10^5^ CFU/mL, which is lower than many authors, who opt for 1.5 × 10^8^ CFU/mL (0.5 on the McFarland scale). The honey proved to have inhibitory and bactericidal effects on all the bacterial strains, with consistent results across species. *E. coli* was both inhibited and killed at 25%, *S. aureus* was inhibited at 12.5% and killed at 25%, while *P. aeruginosa* was inhibited and killed at 25% and 50%, respectively [95]. This study suggests that all three of these ESKAPE pathogens are sensitive to treatment with honey, although a resistance profile of the organisms was not provided by the study. This could have provided valuable information on the potential of honey where antibiotics cannot be used.

Honey has also shown beneficial effects when used as a dressing for DFU patients. A clinical trial performed in Malaysia compared a standard dressing with one that used regular food-grade honey that had not been reported to be sterile [98]. Patients still received antibiotic therapy alongside and tissue specimens were taken during debridement for culture. Analysis of the organisms found in the ulcers prior to treatment showed that 20% of patients had polymicrobial infections populated by organisms including *Streptococcus* spp., *Bacteroides*, *Staphylococcus* spp., *Pseudomonas*, *Enterococcus, Acinetobacter*, and *E. coli*. Only one patient out of 30 had no reported infection. Analysis of the control group showed that while some organisms were culture-negative at the end of treatment, wounds carrying *Pseudomonas*, *Enterococcus*, and those that were polymicrobial were still infected. The group who received honey treatment still had persistent *Enterococcus* and *Bacteroides* at the end of the treatment period. Although some bacteria persisted, the group who received honey-based dressings reported less pain, less oedema, and reduced odour of the wound compared to the control. Neither group reported infection from new organisms [97]. Although there was no noticeable effect on wound healing time, this would have been hard to measure, as the wounds were different sizes to begin with and the study did not use any specific parameters by which to measure [99].

In an earlier case study by Cooper et al. (2001) [98], a patient had a chronic wound that had failed to heal for 3 years after surgery and requested topical treatment with honey. The patient was prescribed co-amoxiclav combined with dressings impregnated with 25–35 g of manuka honey (MGO not reported) with instructions to redress the wound daily. After 7 days of using the honey-impregnated dressing, the wound was smaller and less inflamed. A swab of the wound also found no bacteria present in the wound and the recurrent infections ceased. After one month, the wound had healed sufficiently, but the skin was not pliable and split easily, so the treatment continued for another two months, after which the wound had healed completely. The patient reported reduced pain and discomfort during the treatment period, suggesting that honey has merits besides its potential antimicrobial and regenerative properties [98]. As with the study by Shukrimi et al. (2008) [99], it is not possible to credit the decrease in bacterial infection solely to the honey, as the honey dressings were used in combination with antibiotics. There are of course ethical implications that need to be considered with human trials, though it is important not to compromise the patient’s recovery by denying them treatments that are understood to have an effect, even if the antibiotics can be limited.

There is some controversy as to how appropriate honey treatments are for some patients, particularly for paediatrics. As mentioned above, honey can contain *C. botulinum*, which can lead to wound botulism if wounds become colonized by the bacteria [85]. This can cause paralysis, and children are particularly susceptible. Although there are recorded cases where children as young as 9 and 10 months old were successfully treated with medical grade honey and recovered well [84], there are opposing authors. Mohd Tamrin (2020) [100] presented a case study of a 1-month-old female with an infected umbilical stump who was treated with antibiotics and the wound dressed with medical grade honey. After 5 days, the patient became very ill and wound botulism was suspected based on her symptoms of paralysis. What this paper lacks (and is acknowledged by the author) is that wound botulism was never accurately diagnosed, but the author claimed there was a strong correlation.

*C. botulinum* can resist gamma radiation, and so it is difficult to conclude that the resulting product is bacteria-free even after radiation treatment. The spores can also be spread unevenly throughout the honey, so it is common for false negatives to be returned when trying to identify the spores [100].

## 4. Clinical Studies

PUFAs are under investigation in clinical trials worldwide, and a search of the National Institutes of Health (NIH) database (correct as of May 2024) shows that there are 568 trials using DHA and 581 using EPA [101]. Further refinement of this search shows that only a small fraction of these studies was associated with wound care, though this is perhaps due to the varied results shown in the literature. Researchers will be unwilling to test a treatment as far as clinical trials without better understanding its effects. Most studies focus on the use of fish oil, while only a few investigate EPA and DHA. Conditions using these substances for trials included fatty liver disease, polycystic ovary syndrome, obesity, sepsis, and various cancers including rectal, breast, and gastrointestinal cancer. Although there were listed studies that included resolvins, these were not based on treatments for wounds and were investigating the resolvins naturally produced by the body during medical conditions such as brain injury, gestational diabetes, and chronic pain.

The details provided about the studies also show that a dietary application is by far the most common route of administration to be studied. The remaining studies all use an intravenous application, leaving no studies trialling topical administration. This search shows that this is an emerging area of clinical research and that there are still many areas relating to the use of fish oil that are lacking in depth.

There are comparatively few clinical trials ongoing (as of May 2024) involving manuka honey, as only 32 studies were listed on the clinical trials website [101]. However, the statistics were investigated at separate times, and it is acknowledged that the ongoing pandemic has caused potential issues that have hindered the start of new clinical trials [102], so the comparison is likely biased. Manuka honey is also acknowledged and recommended as an existing treatment in many places [91] and so may be under less investigation as a new treatment, while marine-based fatty acids are not currently a common treatment option.

Of the listed trials involving manuka honey, 40% are already associated with wound healing in many cases, and involve care of second- and third-degree burns, DFUs, postoperative dental care, and use for children with pressure ulcers. Other uses for investigation include eye drops for treatment of conjunctivitis, sinus rinses, and esophagitis pain relief for radiation treatment. While the scope of potential uses for DHA and EPA is wide, manuka honey trials are more limited and mostly cover wound healing and antimicrobial issues.

## 5. Conclusions

Chronic wounds are becoming increasingly prevalent and have a large impact on the patient’s quality of life, as well as the resources required for prolonged treatment. Due to the nature of chronic wounds, bacterial infection is of concern due to disruption of the skin that acts as the protective barrier to infection [2]. Existing work on DHA and EPA includes investigations into dental pathogens [59], where they were effective at much lower concentrations than required for the majority of the ESKAPE pathogens. As these have also been shown to be antibiofilm treatments, this could potentially make them of interest as preventative agents in plaque formation if they demonstrate no increase in resistance. This would require more extensive investigation by widening the bacterial profile that DHA and EPA have to date been tested against, but could aid in prevention of periodontal disease through plaque build-up, which leads to inflammation and tissue destruction [103]. Additionally, some chronic conditions are associated with oral and systemic inflammation leading to increased risk of periodontal disease, such as cardiovascular disease and diabetes mellitus [104].

There is a substantial amount of primary evidence to suggest that manuka honey helps wounds to heal well [98,104], and there is also evidence that application of fish skin to open surface wounds aids in wound healing [103]. Manuka honey has shown to be a potent antimicrobial agent due to its MGO content [79,105], which can in turn lead to a better wound closure time. Other varieties of honey may provide a better effect in accelerating wound closure due to their other properties, such as the high phenol or antioxidant qualities shown in buckwheat or acacia honey [106].

However, the varied methods between the studies highlight the need for a standardised approach to research and accurate reporting of the influencing factors. This includes a more detailed analysis of the manuka honey, at the very least containing the MGO, but the field would benefit from a wider analysis panel of qualities such as the phenol content, sugar content, osmolality, trace metals, and specific gravity. Fish oil also contains other components, including protectins and maresins. It is also understood that EPA and DHA have other metabolites besides resolvins such as leukotrienes [26]. As they can provide a complete bactericidal effect on planktonic bacteria, it should be investigated if this is due to the acid acting as the treatment or whether the resolvins are subsequently being produced in solution, and if so, which ones and at what concentrations. An investigation of other available resolvins would aid a wider-range picture of the antimicrobial activity of each separate resolvin. Another emerging area of resolvin research is the investigation of aspirin-mediated resolvins. It is understood that aspirin can lead to the formation of slightly different aspirin-triggered resolvins [107] and a significant increase in phagocytosis from aspirin-triggered RvD3 when compared to regular RvD3 [32]. These alternative resolvins could also be assessed under more standardised methodology to enable better comparisons.

Although honey is already used in some clinical settings, it may not be a suitable option for some patients, such as young children, diabetic patients, or those with bee allergies (as an immune response could be triggered) [30,97]. Improved identification of the range of the substances within the honey and the mechanism of action to pinpoint the active component(s) with the best regenerative effects, particularly as some research suggests buckwheat and acacia honey may lead to faster wound closure than manuka honey, is needed. Together, this may prove beneficial in research towards synthesis of active molecules from honey or fish oil to provide the desired regenerative or antimicrobial effects and provide suitable treatment options without potential for reaction for a wider range of patients.

## Figures and Tables

**Figure 1 microorganisms-12-01593-f001:**
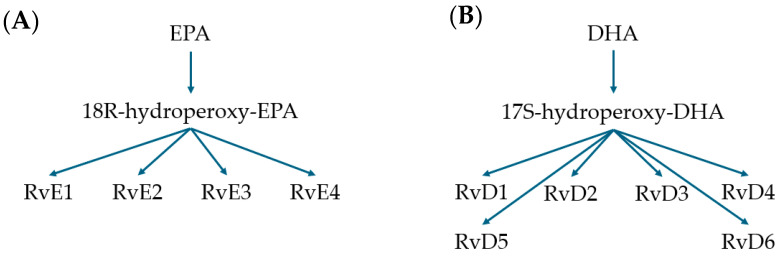
Pathways demonstrating the starting materials of (**A**) eicosapentaenoic acid (EPA) and (**B**) docosahexaenoic acid (DHA) and their intermediates in synthesising resolvins.

**Figure 2 microorganisms-12-01593-f002:**
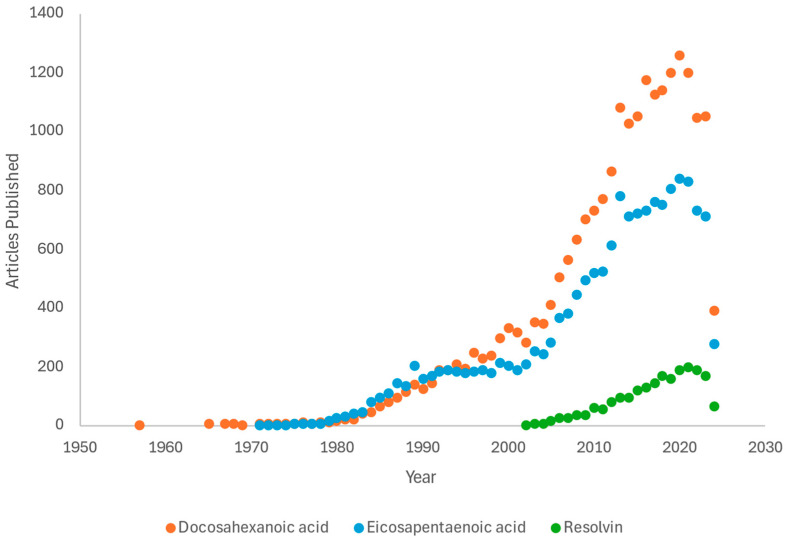
An online search using PubMed for published articles whose titles contain either docosahexaenoic acid, eicosapentaenoic acid, or resolvin based on the year published, showing the increase in published research since the 1980s. Due to other chemicals using similar abbreviations, search terms were restricted to full names only. Data correct as of May 2024.

**Figure 3 microorganisms-12-01593-f003:**
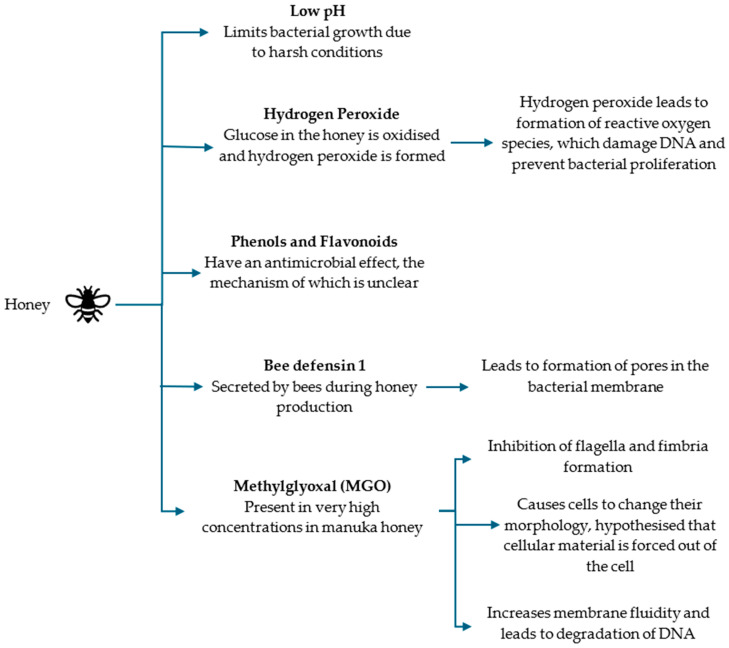
Summary of the ways in which the different components of honey are thought to exhibit antimicrobial effects on bacteria [73,77,78,79,80,81].

## Data Availability

Data are available on request from the corresponding author.

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
