# Peer review of "The Potential of Fish Oil Components and Manuka Honey in Tackling Chronic Wound Treatment"

_microorganisms, 2024, doi:10.3390/microorganisms12081593_

Round 1

Reviewer 1 Report

Comments and Suggestions for Authors

The manuscript presents a review of the antimicrobial and wound-healing properties of manuka honey and fish oil components. The comparison provided here does not offer any significant new information or perspectives that would advance the field in a meaningful way. While the topic of antimicrobial resistance and chronic wound management is undeniably important, the manuscript does not significantly contribute to the discourse. Its lack of originality, coupled with significant presentation shortcomings, renders it of limited value. Nevertheless, the article may be published after a substantial revision of the presentation of already known data and the introduction of original concepts related to the review topic.

Major Comments:

The main issue is the lack of a clear comparison of the common and distinct mechanisms of action, advantages, and disadvantages of fish oil components and manuka honey in tackling chronic wound treatment. I carefully read the manuscript but did not find a sufficient explanation for this intriguing combination of study objects chosen by the authors for the same review. Why were fish oil components and manuka honey selected for consideration instead of, for example, fish oil components and antimicrobial peptides? Or why wasn't manuka honey examined in more detail in comparison with clinically proven wound healing drugs? I believe that additional schematic diagrams and tables, clearly explaining the common and different mechanisms of action of fish oil components and manuka honey, their advantages and disadvantages compared to wound healing drugs and the prevention of bacterial infections, could significantly improve the logic and justification of the review.

Line 144. “Levy et al.” should correctly be cited as “Kohli and Levy” according to the author order in the referenced article. Additionally, after this cited article, the order of references is misaligned, as seen in Lines 724-727, where two articles are combined under reference number 23. Please correct this.

Lines 297-299. The sentence comparing the number of published articles concerning bacteriophages and EPA (eicosapentaenoic acid) is somewhat confusing, as it seems to lack a clear context or connection between the two topics. It's strange that bacteriophages are mentioned only in this sentence without further explaining their significance. To improve coherence and relevance, the sentence should either provide a rationale for comparing bacteriophages and EPA or be rephrased to focus on a more relevant comparison.

Minor comments:

Line 363. “from 107 to 105 CFU” should likely have the numbers 7 and 5 written as superscripts, as they refer to powers of 10.

Line 371. “DFU” is used for the first time, please define it. Since it likely stands for “diabetic foot ulcer,” the abbreviation DFU could be introduced for clarity at its first mention on Line 41.

Line 494-495. “To be considered as a medical grade honey, it must be gamma irradiated so as to rid the honey of bacteria [74].” Are there other methods for producing “medical grade honey”? Are filtration, UV irradiation, or ozone treatment suitable?

Line 528. “ppb” should be spelled out.

Line 534. “DDT, HCH, and HCB” should also be spelled out as they are mentioned for the first time.

Author Response

The manuscript presents a review of the antimicrobial and wound-healing properties of manuka honey and fish oil components. The comparison provided here does not offer any significant new information or perspectives that would advance the field in a meaningful way. While the topic of antimicrobial resistance and chronic wound management is undeniably important, the manuscript does not significantly contribute to the discourse. Its lack of originality, coupled with significant presentation shortcomings, renders it of limited value. Nevertheless, the article may be published after a substantial revision of the presentation of already known data and the introduction of original concepts related to the review topic.

We thank the reviewer for their critique and comments, which we have taken on board and revised the manuscript in accordance with the comments.

Major Comments:

The main issue is the lack of a clear comparison of the common and distinct mechanisms of action, advantages, and disadvantages of fish oil components and manuka honey in tackling chronic wound treatment. I carefully read the manuscript but did not find a sufficient explanation for this intriguing combination of study objects chosen by the authors for the same review. Why were fish oil components and manuka honey selected for consideration instead of, for example, fish oil components and antimicrobial peptides? Or why wasn't manuka honey examined in more detail in comparison with clinically proven wound healing drugs? I believe that additional schematic diagrams and tables, clearly explaining the common and different mechanisms of action of fish oil components and manuka honey, their advantages and disadvantages compared to wound healing drugs and the prevention of bacterial infections, could significantly improve the logic and justification of the review.

The review is focused on the potential of natural substances in tackling chronic wound management, the two prominent study objects fish oil and honey are already produced as food grade ingredients and medical grade products with established manufacturing and global distribution – this has been more clearly explained from the outset including in the review title, abstract (lines 16, 23-26) and introduction (lines 81-83).

The mechanisms of action of fish oil components are not properly defined or well understood, and as such there are no distinct pathways that are suitable for outlining in a figure. The current understanding is discussed between lines 271-296, where the results of some studies have suggested that fatty acids disrupt the cell membrane. Figure 3 has been added to summarise the proposed mechanisms of action of the various components of honey, as there are numerous to consider.

Line 144. “Levy et al.” should correctly be cited as “Kohli and Levy” according to the author order in the referenced article. Additionally, after this cited article, the order of references is misaligned, as seen in Lines 724-727, where two articles are combined under reference number 23. Please correct this.

Thank you for pointing this out, references 18 Kholi and Levy and 23 Levy et al had swapped places in the reference list by mistake, on line 144 the reference was correct Levy et al. 2007 [23] but Ruiz was sitting in the same position in the reference list (lines 727-730) and needed moving into position [24]. Due to the other modifications made to the manuscript this can now be seen on lines 828- 832.

Lines 297-299. The sentence comparing the number of published articles concerning bacteriophages and EPA (eicosapentaenoic acid) is somewhat confusing, as it seems to lack a clear context or connection between the two topics. It's strange that bacteriophages are mentioned only in this sentence without further explaining their significance. To improve coherence and relevance, the sentence should either provide a rationale for comparing bacteriophages and EPA or be rephrased to focus on a more relevant comparison.

To address any confusion, we have provided additional context (lines 324-332) for the comparison between bacteriophage and fish oils as emerging alternatives to antibiotics with greater levels of research for the more targeted bacteriophage compared to the broader spectrum fish oil activity.

Over the past few decades, research has steadily increased in marine-based fatty acids (Fig. 2) but is still low, especially compared to other emerging antimicrobial alternatives to antibiotics such as bacteriophage therapy. Whereas fish oils have the potential for antimicrobial efficacy against multiple pathogenic organisms, bacteriophage are viruses that infect bacteria and bacteriophage therapy uses specific phage viruses that target and kill specific pathogenic bacterial strains. Journal articles concerning bacteriophage have remained high over the past few decades with investigation into therapeutic phage cocktails and non-therapeutic applications, in 2019 there were 1504 papers published concerning bacteriophage technology, and 804 regarding the broader spectrum EPA.

Minor comments:

Line 363. “from 107 to 105 CFU” should likely have the numbers 7 and 5 written as superscripts, as they refer to powers of 10.

Thank you, we have addressed this typographical error, line 396

Line 371. “DFU” is used for the first time, please define it. Since it likely stands for “diabetic foot ulcer,” the abbreviation DFU could be introduced for clarity at its first mention on Line 41.

Thank you, we have introduced the abbreviation at the first mention of diabetic foot ulcers on line 31.

Line 494-495. “To be considered as a medical grade honey, it must be gamma irradiated so as to rid the honey of bacteria [74].” Are there other methods for producing “medical grade honey”? Are filtration, UV irradiation, or ozone treatment suitable?

Thank you, we have added more information for clarity as follows

Line 527-529. To be considered as a medical grade honey, it must be sterilised to rid the honey of bacteria, kill dormant bacterial endospores and follow strict standards and regulations for quality, processing and storage [80].

Lines 544-548. Sterilisation may be achieved through different methods including heat steriliza-tion which is commonly applied to food grade honey and will kill bacteria, but many methods also inactivate the anti-microbial and pro-healing components of honey, whereas gamma irradiation and ozonation are the two sterilization techniques shown to preserve these properties with gamma irradiation providing superior elimination of bacteria [84].

Lines 582-587. There is a lack of harmonized strict guidelines for standardized formulations of medical grade honey and ambiguity around adherence of honey-based wound care products to medical grade standard. Efforts made towards safer medical grade honey formulations effective for wound healing include the development of guidance on standards and criteria for collection, contamination, sterilization, production, storage, physiochemical, legal and safety issues for medical grade honey [84].

Line 528. “ppb” should be spelled out.

Line 534. “DDT, HCH, and HCB” should also be spelled out as they are mentioned for the first time.

Thank you, we have carefully reviewed and revised the manuscript and described all abbreviations including ppb on line 570 and DDT, HCH, HCB on lines 576-7.

Reviewer 2 Report

Comments and Suggestions for Authors

The MS by Clare et al. is a review devoted to the application of fish oil and manuka honey as antibacterial components for wound dressing. Since the antibacterial mechanism of fish oil components is unclear now, the authors provided statistics from each cited work to correctly evaluate the published data. This trend goes on in the subsections devoted to manuka honey. This point highlights the responsibility of the coauthors who created this review.

I have some comments for the authors.

1. The MS contains not enough figures. I recommend adding some schemes of proposed antibacterial action of PUFAs and manuka honey, or their interactions with human cells, as well as some graphical abstracts from cited OA papers.

2. In vitro and in vivo should be italic.

3. There are some non-described abbreviations in the text. For example, COX-2, EUCAST, etc. The coauthors should carefully revise the manuscript.

4. The authors should draw a conclusion section summarizing the state-of-the-art in fish oil and manuka honey applications in present wound healing and highlighting the future prospects of their use.

Author Response

The MS by Clare et al. is a review devoted to the application of fish oil and manuka honey as antibacterial components for wound dressing. Since the antibacterial mechanism of fish oil components is unclear now, the authors provided statistics from each cited work to correctly evaluate the published data. This trend goes on in the subsections devoted to manuka honey. This point highlights the responsibility of the coauthors who created this review.

We thank the reviewer for their summary and comments and provide our responses and revisions below.

I have some comments for the authors.

  1. The MS contains not enough figures. I recommend adding some schemes of proposed antibacterial action of PUFAs and manuka honey, or their interactions with human cells, as well as some graphical abstracts from cited OA papers.

Thank you, this was also raised by reviewer 1 and we have added in figure 3 to provide more graphics.

  1. In vitroand in vivo should be italic.

Thank you, we have italicized in all places throughout the manuscript.

  1. There are some non-described abbreviations in the text. For example, COX-2, EUCAST, etc. The coauthors should carefully revise the manuscript.

Thank you, we have carefully reviewed and revised the manuscript and described all abbreviations including; COX-2, Rv, E. coli, IL, TNF, MRSA, FakAB, FadL, ACP, ACR, CFU, EUCAST, DNA, RNA, MIC, DFU, EU, DDT, HCH, HCB, NIH.

  1. The authors should draw a conclusion section summarizing the state-of-the-art in fish oil and manuka honey applications in present wound healing and highlighting the future prospects of their use.

Thank you, we have added Section 5 Conclusion from Lines 720-767 including future prospects and relevant new references

  1. Conclusion

Chronic wounds are becoming increasingly prevalent and have a large impact on the patient’s quality of life, as well as the resources required for prolonged treatment. Due to the nature of chronic wounds, bacterial infection is of concern due to disruption of the skin which acts as the protective barrier to infection [2]. Existing work around DHA and EPA include investigations surrounding dental pathogens [58], where they were effective at much lower concentrations than required for the majority of the ESKAPE pathogens. As they have also been shown to be antibiofilm treatments, this could potentially make them of interest as preventative agents in plaque formation if they demonstrated no increase in resistance. This would require more extensive inves-tigation by widening the bacterial profile that DHA and EPA have to date been tested against but could aid in prevention of periodontal disease through plaque build-up, which leads to inflammation and tissue destruction [98]. Additionally, some chronic conditions are associated with oral and systemic inflammation leading to increased risk of periodontal disease, such as cardiovascular disease and diabetes mellitus [99].

There is a substantial amount of primary evidence to suggest that manuka honey helps wounds to heal well [94; 100], and there is also evidence that application of fish skin to open surface wounds aids in wound healing [101] manuka honey has shown to be a potent antimicrobial agent due to its MGO content [102-103], which can in turn lead to a better wound closure time, other varieties of honey may provide a better effect in accelerating wound closure due to their other properties, such as the high phenol or antioxidant qualities shown in buckwheat or acacia honey [104].

However, the varied methods between the studies highlight the need for a stand-ardised approach to research and accurate reporting of the influencing factors. This in-cludes a more detailed analysis of the manuka honey, at the very least containing the MGO, but the field would benefit from a wider analysis panel of qualities such as the phenol content, sugar content, osmolality, trace metals and specific gravity. Fish oil also contains other components including protectins and maresins, it is also understood that EPA and DHA have other metabolites besides resolvins such as leukotrienes [26]. As they can provide a complete bactericidal effect on planktonic bacteria it should be in-vestigated if this is due to the acid acting as the treatment or whether the resolvins are subsequently being produced in solution, and if so which ones and at what concentra-tions. An investigation of other available resolvins would aid a wider range picture of the antimicrobial activity of each separate resolvin. Another emerging area of resolvin research is the investigation of aspirin mediated resolvins. It is understood that aspirin can lead to formation of slightly different aspirin triggered resolvins [105] and a signif-icant increase of phagocytosis from aspirin triggered RvD3 when compared to regular RvD3 [32]. These alternative resolvins could also be assessed under more standardised methodology to enable better comparisons.

Although honey is already used in some clinical settings, it may not be a suitable option for some patients such as young children, diabetic patients or those with bee allergies (as an immune response could be triggered) [30; 95]. Improved identification of the range of the substances within the honey and the mechanism of action to pinpoint the active component(s) with the best regenerative effects, particularly as some research suggests buckwheat and acacia honey may lead to faster wound closure than manuka honey. Together, this may prove beneficial in research towards synthesis of active molecules from honey or fish oil to provide the desired regenerative or antimicrobial effects and provide suitable treatment options without potential for reaction for a wider range of patients.

Reviewer 3 Report

Comments and Suggestions for Authors

This review considers how marine based fatty acids (including their metabolites) and manuka  honey can act as antimicrobials, as well as how they can aid in wound repair, two important aspects  leading to resolution of chronic wounds.

 The authors are called to go into detail about all the aspects of wound healing, acute wounds, chronic wounds (not only diabetics) with appropriate and recent .bibliographical entries. This important chapter have to be added and not included in the introduction

Comments on the Quality of English Language

 Moderate editing of English language required

Author Response

This review considers how marine based fatty acids (including their metabolites) and manuka  honey can act as antimicrobials, as well as how they can aid in wound repair, two important aspects  leading to resolution of chronic wounds.

 The authors are called to go into detail about all the aspects of wound healing, acute wounds, chronic wounds (not only diabetics) with appropriate and recent .bibliographical entries. This important chapter have to be added and not included in the introduction

Moderate editing of English language required

We thank the reviewer for their summary and suggestions, we have added a section in the introduction to provide more detail about wound healing in different types of wounds with appropriate and new recent references from lines 30-57 as shown below, and we have carefully reviewed the manuscript and improved the editing and language throughout.

Surface wounds are a common issue and are classified as either acute or chronic, with further sub classifications dividing them into pressure ulcers, diabetic foot ulcers (DFUs), traumatic wounds and surgical wounds [1]. In all these instances the skin is disrupted, breaking open the immune system’s first line of defence and leaving the body vulnerable to pathogens that may enter [2]. Often wounds heal well, but they can be impeded by excessive inflammation and infections. Chronic wounds are also becoming increasingly more common worldwide, with an estimated 15% of over 70’s living with one [3]. These more serious wounds persist for an average of 12-13 months and up to 70% of patients report reoccurrence of the wound [4]. There is a lack of definition as to how long a wound should normally take to heal as too many factors are present in the process including wound size, type and the patient’s age, diet, or other health complications [5]. There are many types of wounds such as surgically induced cuts and burns, but diabetic foot ulcers are a particular example of potentially chronic wounds as they are often impeded by other complications caused by diabetes such as lack of blood flow to the extremities and a lack of growth factors at the site of injury [6]. Due to the in-creasing number of cases of diabetes across Europe and worldwide [7], diabetic foot ulcers are fast becoming a large burden on many health care services and so effective treatments for chronic wounds must be identified.

Wound healing is split into four phases; haemostasis, inflammation, proliferation and remodelling. In haemostasis the body reacts to wound infliction by staunching the blood flow to minimise loss. This is achieved by activation of the clotting cascade via either the intrinsic or extrinsic pathways, or by activating platelets. In all of these sce-nario’s thrombin is activated and a clot will form, either by platelet aggregation or conversion of fibrinogen to fibrin [8]. Inflammation consists of two phases; the initiation and resolution phases. Inflammation can also be split into either chronic or acute cate-gories, where acute inflammation is resolved in good time. If not, acute inflammation becomes chronic [9] and is often linked to a range of conditions including diabetes, asthma, rheumatoid arthritis, Alzheimer’s disease and cardiovascular disease [10].

Round 2

Reviewer 1 Report

Comments and Suggestions for Authors

The authors have made some revisions to the manuscript, which have somewhat improved the original version of this review. I believe that after minor corrections, the manuscript can be published.

Minor comment:

Line 28. It makes sense to add "resolvins" to the list of keywords.

Author Response

Line 28. It makes sense to add "resolvins" to the list of keywords.

Response: Thank you for highlighting this, we have now added resolvins to the keywords at line 27

Reviewer 2 Report

Comments and Suggestions for Authors

The authors made required corrections, so MS can be accepted

Author Response

Thank you for taking the time to review our manuscript

Reviewer 3 Report

Comments and Suggestions for Authors

The authors have asked correctly to my questions.

Comments on the Quality of English Language

 Minor editing of English language required

Author Response

(The authors gave the same response as above.)
